# Where will it end? Pathways to care and catastrophic costs following negative TB evaluation in Uganda

Thomas H. A. Samuels[1,2]*, Priya B. Shete[3,4], Chris Ojok[3,5], Talemwa Nalugwa[3], Katherine Farr[3,4], Stavia Turyahabwe[3,6], Achilles Katamba[3,6,7], Adithya Cattamanchi[3,4], David A. J. Moore[1,2,3]

1 London School of Hygiene and Tropical Medicine, London, United Kingdom, 2 Hospital for Tropical Diseases, University College London Hospitals NHS Foundation Trust, London, United Kingdom, 3 Uganda Tuberculosis Implementation Research Consortium, Kampala, Uganda, 4 Center for Tuberculosis and Division of Pulmonary and Critical Care Medicine, University of California San Francisco, San Francisco, California, United States of America, 5 School of Medicine, Makerere University College of Health Sciences, Kampala, Uganda, 6 National Tuberculosis and Leprosy Programme, Uganda Ministry of Health, Kampala, Uganda, 7 Implementation Science Program, Johns Hopkins Bloomberg School of Public Health, Baltimore, Maryland, United States of America

* thasamuels@gmail.com

**Data Availability Statement:** All relevant data are within the manuscript and its Supporting Information files.

## Abstract

### Introduction

Catastrophic costs incurred by tuberculosis (TB) patients have received considerable attention, however little is known about costs and pathways to care after a negative TB evaluation.

### Materials and methods

We conducted a cross-sectional study of 70 patients with a negative TB evaluation at four community health centres in rural and peri-urban Uganda. Patients were traced 9 months post-evaluation using contact information from TB registers. We collected information on healthcare visits and implemented locally-validated costing questionnaires to assess the financial impact of their symptoms post-evaluation.

### Results

Of 70 participants, 57 (81%) were traced and 53 completed the survey. 31/53 (58%) surveyed participants returned to healthcare facilities post-evaluation, making a median of 2 visits each (interquartile range [IQR] 1–3). 11.3% (95%CI 4.3–23.0) of surveyed patients and 16.1% (95%CI 5.5–33.7%) of those returning to healthcare facilities incurred catastrophic costs (*i.e.*, spent >20% annual household income). Indirect costs related to lost work represented 80% (IQR 32–100%) of total participant costs.

### Conclusions

Patients with TB symptoms who experience financial catastrophe after negative TB evaluation may represent a larger absolute number of patients than those suffering from costs due

**Funding:** DJAM received a grant from the UK Medical Research Council (grant number MR/M017362/1; https://mrc.ukri.org/). AC received a grant from the National Heart, Lung, and Blood Institute (grant number R01HL130192 https://www.nhlbi.nih.gov/). THAS received funding from the Masters Trust Fund at the London School of Hygiene and Tropical Medicine (Project Code ITCR 082010; www.lshtm.ac.uk). The funders had no role in study design, data collection and analysis, decision to publish, or preparation of the manuscript.

**Competing interests:** The authors have declared that no competing interests exist.

to TB. They may not be captured by existing definitions of non-TB catastrophic health expenditure.

## Introduction

Whilst the incidence and mortality of tuberculosis (TB) are declining globally, they are not declining fast enough to meet the ambitious targets set out in the World Health Organization's (WHO) END TB strategy [1]. This strategy highlights the need for bold social policies and research in addition to patient-centred care in order to eliminate TB. To this end, there has been increased interest in the diagnostic journey of TB patients and their financial costs as they represent potential opportunities for novel interventions and programmatic development (1,2).

The severe economic implications of a TB diagnosis in lower and middle income countries have been well described [2, 3]. The proportion of patients incurring catastrophic costs (defined as >20% annual household income (AHI)) whilst obtaining a TB diagnosis and treatment serves as a measure of the financial burden of TB. Eliminating catastrophic costs for TB affected households is one of the three key targets in the END TB strategy [1, 4, 5]. However, much less attention has been paid to those who test negative during TB evaluation. These patients represent the vast majority of those embarking upon the TB diagnostic journey. Even in high TB-burden settings such as Uganda, 80–90% of patients with chronic cough will have a negative evaluation for TB [6–8]. Though they likely do not have the disease, persistent symptoms may lead patients to seek alternative diagnoses and therapies even after their negative TB evaluation. It is currently unclear to what extent these patients are at risk of incurring punitive costs in the pursuit of such a solution.

Studies of TB patients suggest that costs due to lost work represent a considerable proportion of their total financial burden [4] leading the END TB strategy to include lost income and non-medical cost in the calculation of 'catastrophic costs' in TB patients [9]. By contrast, financial risk in non-TB patients with similar respiratory symptomatology only considers out-of-pocket medical costs, termed by WHO as 'catastrophic health expenditure' [10]. It is not known to what extent non-TB patients suffer from non-medical and indirect costs that may not be captured under this current definition.

Previous studies have shown TB-negative individuals are likely to incur catastrophic costs prior to testing when their TB status is unknown [11]. However, to our knowledge, the financial burden accumulated after a negative TB evaluation has not been evaluated. We sought to characterise the number and type of healthcare providers visited by patients after a negative TB evaluation in Uganda and to estimate both the direct and indirect costs they incurred in following these pathways of care.

## Materials and methods

### Ethics statement

Ethical approval was granted by institutional review boards at the London School of Hygiene and Tropical Medicine (London, UK; MSc Ethics Ref: 15360), Makerere College of Health Sciences (Kampala, Uganda; REC Ref 2016–037) and University of California San Francisco (San Francisco, California, USA; ref:221338 IRB# 15–17296). Written consent was obtained for all in person interviews. Oral consent was taken and recorded for all telephone interviews.

## Study setting

This exploratory cross-sectional study was carried out in four geographically-distinct areas of Uganda, two peri-urban and two rural, all within 150km of Kampala. Study sites were selected from those participating in XPEL TB trial, a cluster randomized trial to evaluate the effectiveness and implementation of onsite GeneXpert testing at community health centres [12]. One community-level TB microscopy centre was selected at random per *a priori*-defined area from those participating in the trial. Each of these centres uses sputum smear microscopy as the primary method of diagnosis for TB, tests more than 150 patients a year and refers sputum samples to a district or regional facility for Xpert MTB/Rif testing as part of the Uganda national TB program Xpert referral network.

## Participants

Potential participants were identified from TB laboratory registers routinely kept at the four centres. Adult patients ($\geq$18 years) who could consent and had a negative TB test result at these centres in quarter three and four of 2017 were eligible for inclusion. Individuals with a positive TB test result or who started treatment for TB within 2 weeks of their original evaluation were excluded. A convenience sample of 70 eligible individuals was taken. Preference was shown for individuals with recorded mobile telephone information, but in all other respects sampling was random. Participants were traced approximately 9 months after their negative test (Fig 1). Once traced, participants were given information about the study purpose and procedures and then asked to provide informed consent. If contacted by telephone, participants were read a standardised information and consent script and asked to consent over the phone–this was recorded as an audio data file.

## Surveys

All traced participants found to be alive were surveyed in participants' local languages. Participants found to have died had the approximate date and reported cause of death recorded. Information was collected for up to ten healthcare facility visits made after a participant's negative TB evaluation. This included the number and type of facility and time taken to visit. Cost data was collected for the post-evaluation period using locally-validated questionnaires adapted from the WHO patient costs survey for TB [9]. The adapted tool is presented in the Supplementary information. Costs incurred prior to and during TB evaluation were not evaluated. Participants were asked to estimate the cost of each component part of their visit (e.g., travel expenses, medication) including the value of any lost income. Total direct (out-of-pocket costs, split into medical e.g., medication, and non-medical e.g., transport) and indirect (lost income) costs were calculated. Total indirect costs per participant were calculated by taking a fraction of reported annual personal income based on the reported number of days of work lost due to symptoms post-TB evaluation. Dissavings were defined as the sum of reported borrowing, selling of assets and taking-out of loans. Cost data were collected in Ugandan Shillings (USh) and reported in both United States Dollars ($) and as a proportion of pre-morbid annual household income. During the data collection period the median nominal exchange rate was 3,680USh to $1USD; this was used calculate and report results in USD.

## Data analysis

Patient characteristics, costs and healthcare-seeking behaviours were described using proportions with 95% confidence intervals for dichotomous outcomes and medians with inter-quartile ranges for non-parametric continuous outcomes. Wilcoxon Rank Sum tests were used to

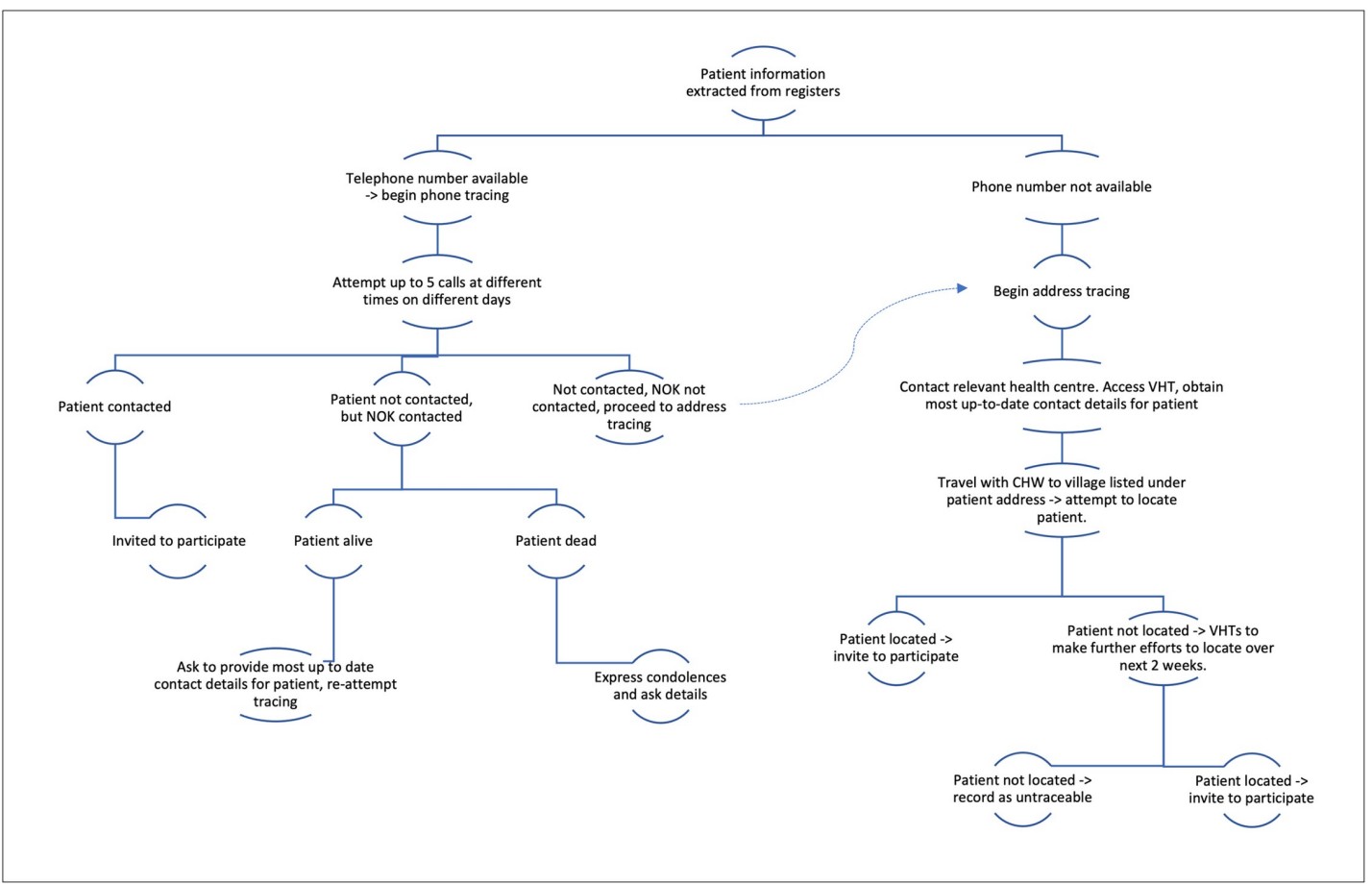

**Fig 1. Participant tracing procedure.** NOK = next of kin; VHT = village health team.

assess statistical differences in time and cost data between dichotomous variables. Univariate and multivariable sub-analyses were conducted using logistic and linear regression for binary and continuous dependent variables respectively. All data were analysed in STATA (version 15, StataCorp USA).

Ethical approval was granted by institutional review boards at the London School of Hygiene and Tropical Medicine (London, UK), Makerere College of Health Sciences (Kampala, Uganda) and University of California San Francisco (San Francisco, California, USA).

## Results

### Participants

Tracing was successful in 57/70 individuals (81%). Four traced individuals had died. Of the remaining 53, all consented to being surveyed. Surveys were administered a median of 10 months following negative TB evaluation (range 8–14 months). The 13 untraced individuals were younger than the 57 traced participants (mean difference -14.5 years 95%CI -23.3 - -5.7). No other statistically significant difference was found between these groups.

The 53 surveyed participants had a median age of 40 (IQR 32–51), 28 (53%) were female (Table 1). Thirty-three were traced by phone (62%). Median AHI (mAHI) was $1,174 (IQR $489–$3,261). Eighty-seven percent (n = 46) visited another healthcare facility before presenting for TB evaluation, making a median of 3 visits (IQR 2–5) pre-evaluation.

**Table 1. Characteristics of surveyed participants.**

| Patient characteristics | | Number/median (percentage/IQR) |
|---|---|---|
| Age (years) | | 40 (32–51) |
| Sex | Male | 25 (47) |
| | Female | 28 (53) |
| HIV status | Positive | 27 (51) |
| | Negative | 24 (45) |
| | Unknown | 2 (4) |
| Living Environment | Rural | 29 (55) |
| | Urban-/Peri-urban | 24 (45) |
| Mobile phone ownership | Yes | 35 (66) |
| | No | 18 (34) |
| Tracing method | Mobile phone | 33 (62) |
| | Address | 20 (38) |
| Test type used in the initial negative TB evaluation | Smear Microscopy | 47 (89) |
| | Xpert MTB/Rif | 6 (11) |
| Awareness of negative TB test result | Aware | 38 (72) |
| | Not aware | 15 (28) |

Total number of surveyed participants = 53

Most participants were evaluated for TB with smear microscopy (n = 47, 89%); the remainder were evaluated with Xpert MTB/Rif. Twenty-eight percent were not aware of the result of their evaluation. Six participants (11.3%; 95%CI 4–23) reported being diagnosed with TB more than 2 weeks after their negative index evaluation and had subsequently been started on anti-tuberculous therapy.

## Pathways to care

Symptoms persisted for a median of 4 weeks post-evaluation (IQR 3–13) in surveyed individuals (n = 53), causing a median loss of 7 days of work (IQR 1–30). Twenty-two participants made no further visit to any health facility post-evaluation. The remaining 31 participants (58%) made a total of 83 visits to healthcare post-evaluation, a median of 2 visits each (IQR 1–3) (Fig 2). Fig 3 shows a flowchart detailing pathways to care.

Participants spent a median of 20 minutes travelling (IQR 10–60) and 2 hours visiting (IQR 1.25–5) per facility visit. The most commonly visited facilities were private clinics or hospitals, constituting 36% of all visits made. Local pharmacies (22%) and level III or IV government health centres (23%) were also commonly visited.

## Costs of healthcare seeking behaviour post-evaluation

Participants (n = 53) incurred a median of $16.46 total costs post-TB evaluation (IQR $7.07-$93.33). Direct and indirect costs represented 20% (IQR 0–68%) and 80% (IQR 32–100%) of total costs respectively. Participants lost a median of $13.17 in indirect costs (IQR $1.05-$65.84). For those who accessed healthcare post-evaluation (n = 31), direct costs were a median of $8.97 (IQR $3.40-$30.57). Out-of-pocket medical costs represented 68% (IQR 59–86%) of total direct costs.

The median cost of a facility visit was $10.19 (IQR $4.35-$28.53) (Table 2). Of the 83 visits made, medical costs occurred in 62 visits and non-medical costs in 72 visits. Purchase of

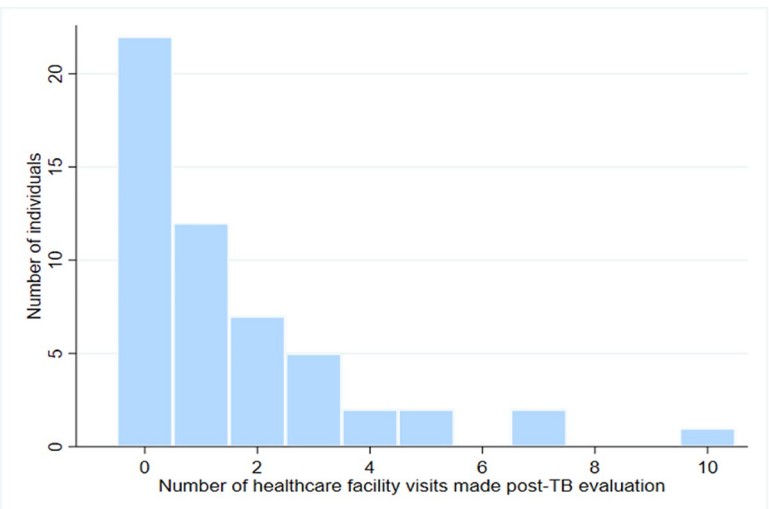

**Fig 2. The distribution of healthcare facility visits made by study participants after TB evaluation.**

medicines was the most frequent medical cost, occurring in 56 (67%) visits. In 38 visits these costs represented the only medical cost. Radiology was the most expensive type of medical cost with a median cost of $6.79. However, it was only accessed in 6 of 83 (7%) visits. Travel costs were the most common type of non-medical direct cost, incurred in 64 of 83 (77%) visits. Food costs were incurred in 32/83 visits and represented a median of 33% (IQR 24–68%) of non-medical costs when they occurred. The fraction of total direct participant expenditure represented by different cost categories is presented in Fig 4.

In total, 11% (95%CI 4.3–23.0%) of study participants spent >20% of AHI post-TB evaluation, incurring catastrophic costs. This amounted to 16.1% (95%CI 5.5–33.7%) of the 31 patients who accessed healthcare post-evaluation. Total financial losses were not significantly greater in those who attended healthcare post-evaluation, although they trended towards being so (median $40.61 [attended] vs $13.64 [did not attend]; p = 0.09). All participants suffering catastrophic costs experienced symptoms for more than 6 weeks post-evaluation.

Of surveyed individuals (n = 53), 66% (95%CI 51.7–78.5%) suffered dissavings post-evaluation. They incurred a median loss of $32.61 (IQR $10.87-$206.52), equivalent to 1.8% of AHI. Borrowing money was the most common form of dissaving (21/53) whilst selling assets incurred the largest median loss ($70.65, IQR $21.74-$353.26). Dissavings were significantly more common amongst participants who attended care post-evaluation (p = 0.04). However, the magnitude of dissaving did not vary between those who attended care post-evaluation and those who did not (p = 0.6). Catastrophic costs trended towards being more common in those that incurred dissavings (p = 0.07).

## Discussion

Patients with chronic cough whose evaluation for TB is negative, although considered to be 'completed' from the perspective of National TB Programmes, remain patients with ongoing symptoms sufficiently troubling as to prompt further healthcare-seeking. More than 10% of individuals experienced financial catastrophe after negative TB evaluation without even considering costs incurred prior to or during TB evaluation. In TB patients these pre-diagnostic costs represent 48–53% of total expenditure [4]. Spending of this magnitude is sufficiently prevalent to warrant the attention of primary healthcare programs.

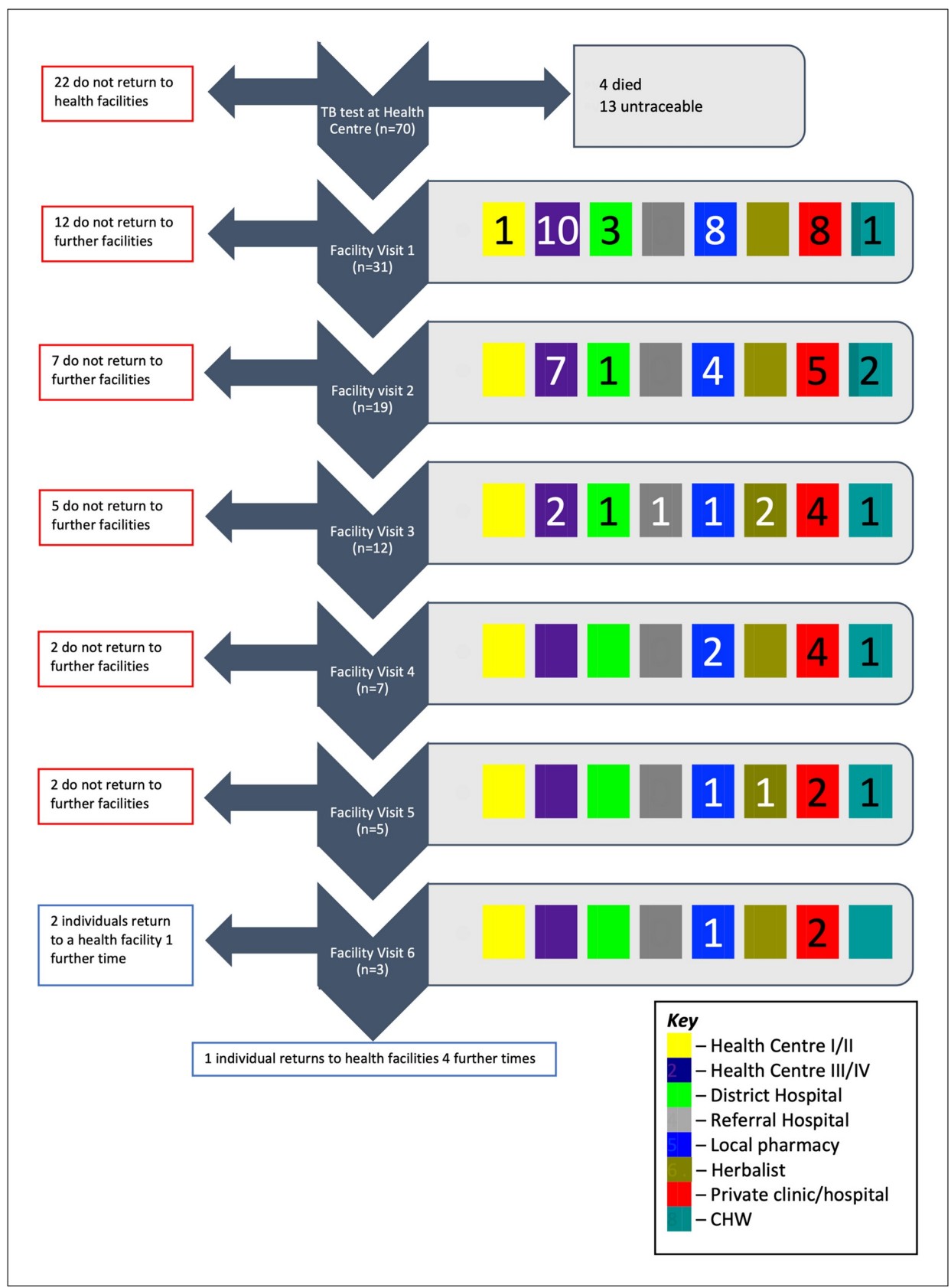

**Fig 3. Patient pathways to care after TB evaluation.** Flowchart showing patient pathways to care after TB evaluation. To the right of each facility visit, coloured boxes represent the types of facilities participants visited (see Key). Numbers within the boxes represent the number of participants visiting that type of facility. Blank boxes indicate that facility type was not visited. If a participant visited a further facility, their next visit is recorded at the next step of the flow chart. If a participant did not visit a further facility, they exit the flow chart to the left of their current facility visit. CHW = Community Health Worker.

Despite a negative evaluation for TB, many individuals continue to search for diagnostic and therapeutic solutions. Although a significant minority did not visit any further facility, more than half did so, some many times over. These results are consistent with the likely range of aetiologies of chronic cough in this setting [13]. The sickest and hardest to diagnose patients may repeatedly access care without resolution, whilst those with self-limiting infectious illnesses require little, if any, further medical attention. However, a lack of accurate diagnostic data means this interpretation warrants further study. Visits to the private sector predominated, contributing 61% of all post-evaluation healthcare episodes. This is consistent with other studies of pathways to care in high-TB burden settings [14, 15].

Many participants experienced significant financial burden from their symptoms post-evaluation, and 11% financial catastrophe. Although this study did not include pre-evaluation costs, the authors and others have previously documented the potential for these costs to be financially catastrophic [4, 11]. Our study shows that for some, these costs do not stop after a negative test for TB. Taken together, the proportion of individuals with non-tuberculous chronic cough experiencing financial catastrophe during their diagnostic journey is likely higher than reported here. TB-negative individuals represent the majority of patients undergoing evaluation for TB. At a community level they could constitute as large a total financial burden as those with the disease. This cost burden likely reflects not only unwell and difficult-to-diagnose individuals who repeatedly access care but also the high indirect cost of seeking healthcare in this environment relative to income. The high burden of indirect costs in this study mirrors those found to occur in TB-affected households, which are characterised by lost income, inability to work and job loss [4, 16–19]. The fact that 28% of participants did not return to receive the results of their TB test is further suggestion that accessing healthcare in this environment is prohibitively expensive for some. The cost of accessing healthcare in lower- and middle-income countries is known to be financially challenging for many and often leads to dissaving and other coping strategies not traditionally assessed in financial risk protection research [20].

There were several limitations to this study. Firstly, the methodology predisposes costing information to recall bias. Steps were taken to mitigate this by making survey questions as unambiguous as possible. Nevertheless, cost findings need to be interpreted with a degree of caution. Second, the study's cross-sectional design does not allow for definitive conclusions to be drawn on causality between observed associations. Last, this study did not include costs incurred prior to evaluation, which limits the extent to which firm conclusions can be drawn

**Table 2. Participant costs per facility visit.**

| | Total cost per facility visit (IQR) | Direct cost per facility visit (IQR) | Direct medical costs per facility visit (IQR) | Direct non-medical costs per facility visit (IQR) | Indirect cost per facility visit (IQR) |
|---|---|---|---|---|---|
| Median cost | 10.19 | 6.79 | 4.35 | 1.36 | 2.72 |
| ($ USD) | (4.35–28.53) | (2.58–16.30) | (0–11.41) | (0.54–4.07) | (0–13.59) |
| Median cost as a percentage of mAHI (%) | 0.9% | 0.6% | 0.4% | 0.1% | 0.2% |
| | (0.4–2.4) | (0.2–1.4) | (0–1.0) | (0.04–0.3) | (0–1.2) |

Costs per facility visit in absolute terms and as a percentage of median annual household income; n = 83. USD = United States Dollar, mAHI = median annual household income, IQR = interquartile range.

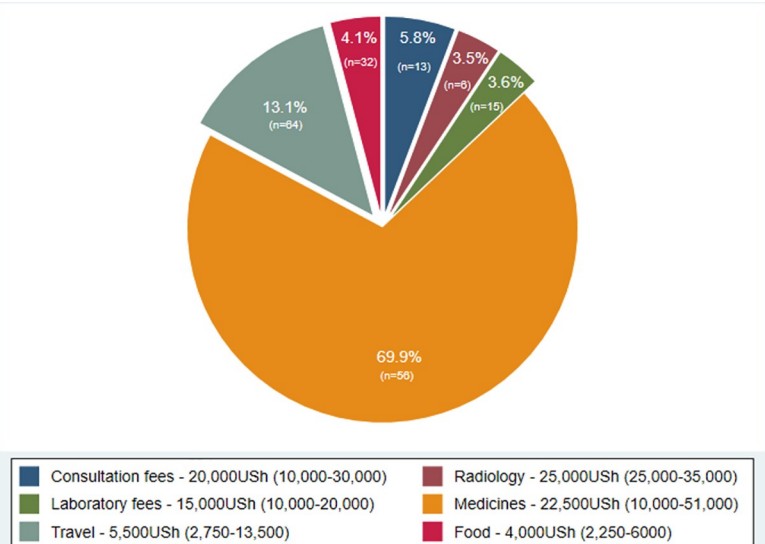

**Fig 4. Distribution of direct costs during facility visits by study participants.** Direct costs incurred in 83 facility visits by the 31 study participants who returned to healthcare after TB evaluation. Slices represent the percentage of overall direct costs incurred by study participants by cost category (percentages labelled). The number of visits in which the specified cost category was accessed is also shown on the slices in brackets. The inner legend shows the median and inter-quartile range (in brackets) spent in that cost category, when accessed.

regarding the total cost incurred by participants during their illness. Despite these limitations, this study focuses on a patient group that has been comparatively neglected by international health research and presents important insights into the financial implications of their health issues. The generalisability of our findings could be seen as a limitation as we purposefully did not institute complex sampling procedures to make our subjects completely representative of the population they are derived from. However, the aim of the study was to describe the situation in these areas of Uganda in order to illustrate issues that could be widespread, rather than to derive specific generalisable conclusions. Of note, rates of phone ownership in study participants are consistent with those of the general population in Uganda [21, 22].

The move towards universal health coverage (UHC), advocated by the WHO as a possible solution to catastrophic health expenditure, may help patients surmount the financial barriers to medical costs for non-TB illnesses. However, UHC-related interventions by themselves will likely not be sufficient to mitigate financial catastrophe for patients as medical costs are only one driver of total patient cost [4, 18, 23]. Parallel social protection initiatives to prevent or reduce non-medical and indirect costs are likely to be essential as these costs make up a significant proportion of patient expenditure. These initiatives have been proven to mitigate such costs in TB patients [24]. Furthermore, under the current definition of catastrophic health expenditure advocated by the WHO, these non-medical and indirect costs are excluded when assessing the financial stress on non-TB patients [10]. A definition of financial catastrophe that accounts for the important financial burdens that non-direct medical costs and indirect costs create would support identification of affected patients and allow health programs to better select and target interventions.

## Conclusions

Far from avoiding the catastrophic costs incurred by many of their TB-positive counterparts, some individuals continue to experience financial catastrophe after negative TB evaluation.

They may represent a larger economic burden at a community level than those suffering from similar costs due to TB and may not be captured by existing definitions of non-TB catastrophic health expenditure. A prospective longitudinal study with recruitment at the point of TB evaluation would allow more accurate cost estimation, capture of final diagnoses and allow exploration of alternative definitions of financial catastrophe and their relationship to outcomes. Furthermore, such a study could be designed to prospectively compare the costs after both a positive and negative TB evaluation from the same population. Meanwhile, more attention needs to be paid to encouraging TB-negative patients to return to the clinic that tested them to coordinate further investigative efforts.

## Supporting information

**S1 Table. Multivariable logistic regression of factors associated with increased odds of attending healthcare facilities after negative TB evaluation.** n = 51 for analysis (due to n = 51 known HIV status). OR = odds ratio; TB = tuberculosis. All variables found to be associated with accessing healthcare post-evaluation with a significance level of $p \leq 0.1$ in the univariate analysis were included in the multivariable analysis along with age and sex. A longer duration of symptoms after TB evaluation was associated with increased odds of attending healthcare facilities post-evaluation. Subjects living in rural settings and those with HIV infection had lower odds. Four or more healthcare attendances prior to TB evaluation trended towards association with increased odds of attendance post-evaluation but was non-significant.
(PDF)

**S2 Table. Univariate logistic regression analysis comparing the unadjusted odds of attending healthcare facilities by study participants after negative TB evaluation.** n = 51 for analysis (due to n = 51 known HIV status). OR = odds ratio; TB = tuberculosis; NTLP = National Tuberculosis and Leprosy Program. Variables that demonstrated an association with a significance level of $p \leq 0.1$ were taken through to multivariable analysis along with age and sex.
(PDF)

**S3 Table. Univariate linear regression analysis comparing the unadjusted total cost incurred by participants after negative TB evaluation.** n = 51 for analysis (due to n = 51 known HIV status). TB = tuberculosis; USh = Ugandan Shillings; NTLP = National Tuberculosis and Leprosy Program.
(PDF)

**S1 File. Modified costing tool.** This tool was adapted from a validated costing tool used in previous work by our group [11]. No questions related to financial expenditure were added or subtracted from the original costing tool. Additional questions were added to aid in discriminating different pathways to care used by study participants.
(PDF)

**S2 File. Raw data for analysis.**
(XLSX)

## Acknowledgments

The authors would particularly like to thank Damalie Nakkonde for her work in the field collecting and screening the data and assisting in translation of questionnaires.

## Author Contributions

**Conceptualization:** Thomas H. A. Samuels, Priya B. Shete, Adithya Cattamanchi, David A. J. Moore.

**Data curation:** Thomas H. A. Samuels, Chris Ojok, Talemwa Nalugwa.

**Formal analysis:** Thomas H. A. Samuels.

**Funding acquisition:** David A. J. Moore.

**Investigation:** Thomas H. A. Samuels, Priya B. Shete, Chris Ojok, Katherine Farr.

**Methodology:** Thomas H. A. Samuels, Priya B. Shete, Adithya Cattamanchi, David A. J. Moore.

**Project administration:** Thomas H. A. Samuels, Chris Ojok, Talemwa Nalugwa, Katherine Farr, Stavia Turyahabwe, Achilles Katamba, Adithya Cattamanchi.

**Resources:** Thomas H. A. Samuels, Talemwa Nalugwa, Katherine Farr, Stavia Turyahabwe, Achilles Katamba.

**Supervision:** Achilles Katamba, Adithya Cattamanchi, David A. J. Moore.

**Writing – original draft:** Thomas H. A. Samuels.

**Writing – review & editing:** Thomas H. A. Samuels, Priya B. Shete, Achilles Katamba, Adithya Cattamanchi, David A. J. Moore.

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
