## [Decision Letter · Decision Letter 0]

11 May 2021

PONE-D-21-07641

Where will it end? Pathways to care and catastrophic costs following negative TB evaluation in Uganda

PLOS ONE

Dear Thomas Henry Ashleigh Samuels, 

Thank you for submitting your manuscript to PLOS ONE. After careful consideration, we feel that it has merit but does not fully meet PLOS ONE’s publication criteria as it currently stands. Therefore, we invite you to submit a revised version of the manuscript that addresses the points raised during the review process.

We ask you to address the minor comments raised by the peer-reviewers including comments on sampling and handling data for example. Please submit your revised manuscript by 10 June 2021. If you will need more time than this to complete your revisions, please reply to this message or contact the journal office at plosone@plos.org. Please include the following items when submitting your revised manuscript:

We look forward to receiving your revised manuscript.

Kind regards,

Eleanor Ochodo, M.D., PhD

Academic Editor

PLOS ONE

Journal Requirements:

Reviewers' comments:

Reviewer's Responses to Questions

**Comments to the Author**

1. Is the manuscript technically sound, and do the data support the conclusions?

Reviewer #1: Yes

Reviewer #2: Yes

2. Has the statistical analysis been performed appropriately and rigorously? 

Reviewer #1: Yes

Reviewer #2: Yes

3. Have the authors made all data underlying the findings in their manuscript fully available?

Reviewer #1: Yes

Reviewer #2: Yes

4. Is the manuscript presented in an intelligible fashion and written in standard English?

Reviewer #1: Yes

Reviewer #2: Yes

5. Review Comments to the Author

Reviewer #1: Summary of the research

The authors set out to characterize the number and type of health care providers visited by patients after a negative TB evaluation. In addition, they sought to estimate both the direct and indirect costs these patents incurred in following these pathways of care. From the participants they surveyed, the authors found that those with negative TB evaluation experience both direct and indirect costs which could be more than what those with positive TB evaluation experience. The findings showed that indirect costs accounted for most of the costs incurred. They found that more than half of the participants surveyed visited health facilities especially private ones after negative TB evaluation with some of them experiencing catastrophic costs especially those with persistent symptoms. These findings show that once someone has negative TB evaluation, that should not be the end of having contact with the person but subsequent follow up on them would be beneficial to enable them to establish the diagnosis and may help minimize the costs. The findings have also highlighted a new area which has received very little focus yet an important area to help achieve universal health coverage in low- and middle-income settings.

Strengths and limitations of the study

The manuscript is well written, easy to follow, and the findings presented clearly. The Participant tracing, the survey and ethical issues have been described clearly. The authors have acknowledged their drawbacks in sampling of the participants, recall bias, study design and generalizability of the findings. They have also highlighted they were not able to do pre-evaluation costs.

Just some minor comments as much as the authors did a convenience sampling for the participants, it is not clear how the four facilities were selected from four geographical areas of Uganda given that there are other facilities in those areas. In addition, for the six participants who reported being diagnosed with TB more than two weeks after their negative index evaluation and were on anti-tuberculous therapy, how was their data handled. More so, two of their references appear to be incomplete.

Examples and Evidence

Major Issues

No major issues

Minor issues

1. In line 87-88,” One community-level TB microscopy Centre was selected per area” can this be described further how the selection was done for instance random, systematically, convenience etc.

2. In figure 1, is it possible to key in the numbers in the participant tracing procedure to show how you ended up with the numbers you surveyed.

In the same figure please give the key for the abbreviations ‘NOK’ and ‘VHT’

3. In Table 1, please state the denominator for the proportions calculated and the units for age.

4. In line 158-160, the six participants who later turned to be positive and were on anti-TB drugs, how did you handle their data?

(i) Were they part of the 31 who visited health care post-evaluation?

(ii) If they were part of the 31, won’t their costs be different from the others who were not on anti-TB drugs?

5. In figure 4, in line 208-209 you have stated what the slices represented, which data did you use to calculate the proportions? One can easily get mixed up with proportions described in line 194 to 199.

6. In line 330, reference 7 and line 372, reference 21 they appear to be incomplete. Are they journal articles? not clear what type of references they are.

7. One of the discussion points, given that majority of those that visited health facilities, visited private health facilities could it explain the high direct medical costs per facility.

8. A suggestion in your recommendations in the conclusion: It would be helpful to do a comparative study between the costs encountered by those having positive TB evaluation and negative TB evaluation in the same population to know the magnitude in both groups.

Reviewer #2: 1. Authors have mentioned it in the limitation, but collecting information on cost incurred over past 9-10 months may have serious recall issues. Have authors tried to access some documents to verify at least the resource use?

2. I would like see more details of indirect cost estimation. How was it estimated if someone was not employed? Was it just by asking how much income they lost for healthcare faculty visit?

3. Figure 2 may not be required and can be explained in text only.

6. PLOS authors have the option to publish the peer review history of their article (what does this mean?). If published, this will include your full peer review and any attached files.

Reviewer #1: No

Reviewer #2: **Yes: **Rakesh Kumar

---

## [Author Response · Author response to Decision Letter 0]

28 May 2021

For ease of reference, the specific reviewer and editor comments and our associated replies are detailed in a table format in the document uploaded as a response to the reviewers. For ease of access, these are listed here too. 

We will address reviewers points below sequentially as raised. 

Reviewer 1:

Comment: Line 87-88: One community-level TB microscopy Centre was selected per area” can this be described further how the selection was done for instance random, systematically, convenience etc.

Response: The centres were selected from amongst those used in a trial to assess the efficacy of on-site GeneXpert testing in local community health centres in Uganda - https://doi.org/10.1186/s13012-020-00988-y

One centre was selected at random from each of the four pre-defined areas from those participating in the above trial. These areas were selected a priori to illustrate a wider variety of patient experience. 

We have amended the methods section to make this clearer

Comment: In figure 1, is it possible to key in the numbers in the participant tracing procedure to show how you ended up with the numbers you surveyed.

In the same figure please give the key for the abbreviations ‘NOK’ and ‘VHT’

Response: Unfortunately, we do not have any more granular data than whether participants were traced by phone, in person, or not at all. This data is available from Table 1. We agree with the authors points about the abbreviations and have amended the legend of the figure. 

Comment: In Table 1, please state the denominator for the proportions calculated and the units for age.

Response: We agree and have made the changes suggested by the reviewer.

Comment: In line 158-160, the six participants who later turned to be positive and were on anti-TB drugs, how did you handle their data?

(i) Were they part of the 31 who visited health care post-evaluation?

(ii) If they were part of the 31, won’t their costs be different from the others who were not on anti-TB drugs?

Response: In response to the reviewer:

1) They were part of the 31.

2) If diagnosed with TB, the subsequent healthcare visits after the visit in which that diagnosis was made were not included in this analysis as for our purposes, they constituted a completed healthcare pathway. TB medication itself is free in Uganda as it is provided by the state, but analysis beyond the point of diagnosis would have included further costs from clinic visits etc. that went beyond the scope of the study.

Comment: In figure 4, in line 208-209 you have stated what the slices represented, which data did you use to calculate the proportions? One can easily get mixed up with proportions described in line 194 to 199.

Response: In figure 4, all direct costs incurred by participants were summed, and then broken down by the type of expenditure as shown. The fraction of overall total costs represented by each type of expenditure was then calculated and is presented as the pie chart in figure 4. 

We agree with the reviewer that the reference to Figure 4 in the main text is misleading in this regard as it refers to different data. We have corrected this. 

Comment: In line 330, reference 7 and line 372, reference 21 they appear to be incomplete. Are they journal articles? not clear what type of references they are.

Response: It appears our referencing software had not fully completed these two references. We have corrected the issue and thank the reviewer for bringing this to our attention. 

Comment: Discussion - One of the discussion points, given that majority of those that visited health facilities, visited private health facilities could it explain the high direct medical costs per facility.

Response: The reviewer makes an excellent point. In data that we have not reported in the manuscript, we found that visits to private facilities had a higher median cost than state facilities (59,000Ush for private facilities vs. 37,000Ush for a district hospital and 27,000Ush for a local health centre). However, perhaps due to the small number of participants in the study, this difference was not statistically significant. Other studies done similar populations tend to support this point. 

This is unlikely to only factor at play, however. In Ugandan state-funded health facilities, medical services beyond a simple consultation, such as radiographs and medications, are often not free at the point of use and health insurance is non-existent. The reported direct medical costs are therefore likely due to more than just the type of facility attended and likely reflect the generally high out-of-pocket expenditure required in this health system.

We chose not to include these points within the discussion as there were other things we wished to explore in more detail.

Comment: Conclusions - A suggestion in your recommendations in the conclusion: It would be helpful to do a comparative study between the costs encountered by those having positive TB evaluation and negative TB evaluation in the same population to know the magnitude in both groups.

Response: We completely agree with the reviewer and have added this suggestion to the conclusion.

Reviewer 2:

Comment: Authors have mentioned it in the limitation but collecting information on cost incurred over past 9-10 months may have serious recall issues. Have authors tried to access some documents to verify at least the resource use?

Response: We agree with the reviewer that these sorts of documents would assist in validating some of the cost results to some extent. Unfortunately, in the vast majority of cases very few such documents exist, and we were not able to collect any within the limits of the study. Ideally, further prospective research needs to occur to validate cost data, as we suggest in the conclusion.

Comment: I would like see more details of indirect cost estimation. How was it estimated if someone was not employed? Was it just by asking how much income they lost for healthcare faculty visit?

Response: Indirect cost estimation was calculated as the fraction of annual reported income lost based on the number of days of work reported lost due to symptoms after TB evaluation/number of total work-days per annum. Separate data was collected asking participants how much lost income occurred due to individual healthcare visits. This was not added to the total calculated above but reported separately as per the results section. If a participant was not employed (almost always due to subsistence farming being their primary economic activity), they were asked to estimate the cost of their lost time. We also attempted to capture the economic costs to these participants by reporting dissavings in the results. 

We have amended the methods to further detail how indirect cost data was calculated

Comment: Figure 2 may not be required and can be explained in text only.

Response: We agree with the reviewer that Figure 2 can be explained in text alone. However, we feel that the simplicity and clarity of the figure is a strength. Given that it remains within the overall figure limit of the paper, we would ideally prefer to keep it in the manuscript.

---

## [Decision Letter · Decision Letter 1]

16 Jun 2021

Where will it end? Pathways to care and catastrophic costs following negative TB evaluation in Uganda

PONE-D-21-07641R1

Dear Thomas Samuels,

We’re pleased to inform you that your manuscript has been judged scientifically suitable for publication and will be formally accepted for publication once it meets all outstanding technical requirements.

Kind regards,

Eleanor Ochodo

Academic Editor

PLOS ONE

Reviewers' comments:

Reviewer's Responses to Questions

**Comments to the Author**

1. If the authors have adequately addressed your comments raised in a previous round of review and you feel that this manuscript is now acceptable for publication, you may indicate that here to bypass the “Comments to the Author” section, enter your conflict of interest statement in the “Confidential to Editor” section, and submit your "Accept" recommendation.

Reviewer #1: All comments have been addressed

Reviewer #2: All comments have been addressed

2. Is the manuscript technically sound, and do the data support the conclusions?

Reviewer #1: Yes

Reviewer #2: Yes

3. Has the statistical analysis been performed appropriately and rigorously? 

Reviewer #1: Yes

Reviewer #2: Yes

4. Have the authors made all data underlying the findings in their manuscript fully available?

Reviewer #1: Yes

Reviewer #2: Yes

5. Is the manuscript presented in an intelligible fashion and written in standard English?

Reviewer #1: Yes

Reviewer #2: Yes

6. Review Comments to the Author

Reviewer #1: The authors have responded to the comments of the reviewer and made the necessary changes in the manuscript. The manuscript has been written well and the research is scientifically sound. The analysis has been done well and the findings have been presented clearly in text ,figures and tables. The authors have drawn their conclusions from their findings.

Reviewer #2: (No Response)

7. PLOS authors have the option to publish the peer review history of their article (what does this mean?). If published, this will include your full peer review and any attached files.

Reviewer #1: No

Reviewer #2: **Yes: **Rakesh Kumar

---

## [Editor Report · Acceptance letter]

7 Jul 2021

PONE-D-21-07641R1 

Where will it end? Pathways to care and catastrophic costs following negative TB evaluation in Uganda. 

Dear Dr. Samuels:

I'm pleased to inform you that your manuscript has been deemed suitable for publication in PLOS ONE. Congratulations! Your manuscript is now with our production department. 

Kind regards, 

on behalf of

Dr. Eleanor Ochodo 

Academic Editor

PLOS ONE